# Culturing experiments reveal mechanisms of daily trace element incorporation into *Tridacna* shells

Authors: Iris Arndt[1,2], Jonathan Erez[3], David Evans[4], Tobias Erhardt[1,2], Adam Levi[3], Wolfgang Müller[1,2]

[1]Institute of Geosciences, Goethe University Frankfurt, Frankfurt am Main, 60316, Germany
[2]Frankfurt Isotope and Element Research Center (FIERCE), Goethe University Frankfurt, Frankfurt am Main, 60316, Germany
[3] Fredy & Nadine Herrmann Institute of Earth Sciences, Hebrew University of Jerusalem, Jerusalem, 9190401, Israel
[4]School of Ocean and Earth Science, University of Southampton, Southampton, SO171BJ, United Kingdom

*Correspondence to*: Iris Arndt (arndt@em.uni-frankfurt.de)

## Abstract

Giant clams such as *Tridacna* are exceptionally well suited for studying past environmental changes on daily to multidecadal timescales. The visible growth bands in their shells, which can be yearly, seasonal or even daily, are accompanied by changes in the elemental composition of the shell and provide insights into their growth and environmental history. The daily elemental cycles, particularly in Mg/Ca and Sr/Ca, can be used to determine age and growth rates. However, the mechanisms creating the visible day and night banding and the associated elemental cycles, remain unclear. To better understand the mechanisms of El/Ca incorporation into the shells of *Tridacna* during day and night growth, we performed controlled growth experiments using $^{135}$Ba-labelled seawater. The isotope spike was alternatingly applied in 12-hour intervals in order to individually and unequivocally mark day and night growth segments in *Tridacna*. These experiments show that *Tridacna* calcification rates are nearly five times higher during the day than at night. In addition, based on the observed changes in shell composition we deduce that the bivalve's extrapallial fluid (EPF) reacts to changes in seawater chemistry within tens of minutes, both during day and night. A full compositional replenishment is achieved after approximately one day, assuming a similar residence time for all elements. During daytime, El/Ca (for El = B, Mg, Sr, Ba) decrease, while Na/Ca increases. The opposite behaviour occurs at night. The night peak in El/Ca occurs in the earliest morning, shortly before the change between spiked and non-spiked water at 7:30. Daily El/Ca cycles are likely dominantly driven by variations in active $Ca^{2+}$ and $HCO_3^-$ transport into the EPF, influenced by light availability, circadian rhythms and/or energy availability (from both photosymbionts and filter feeding), rather than a closed-system Rayleigh fractionation process driven by contrasting El-distribution coefficients alone. We propose that active $Ca^{2+}$ and $HCO_3^-$ pumping into the EPF might also drive diurnal changes of growth rate, shell structure and possibly organic content.

## 1 Introduction

Giant clams serve as archives recording palaeoenvironmental changes in (sub)tropical reefs since their emergence in the middle Eocene, with *Tridacna* emerging in the early Miocene (Harzhauser et al., 2008). Their large, dense aragonitic shells are less prone to diagenetic recrystallisation compared to other reef organisms such as corals (Griffiths et al., 2013; Veeh and Chappell, 1970; Welsh et al., 2011). In addition, *Tridacna* clams live for several decades (Arndt et al., 2025; Rosewater, 1965; Watanabe et al., 2004) and grow quickly, at rates ranging from

millimetres to centimetres per year (Arndt et al., 2023; Bonham, 1965; Elliot et al., 2009; Fousiya et al., 2024; Fursman et al., 2025; Ma et al., 2020; Mills et al., 2023; Rosewater, 1965; Warter et al., 2018). This combination of excellent preservation potential and rapid growth makes them valuable for reconstructing past environmental changes on daily to multiannual timescales (Warter et al., 2015; Warter and Müller, 2017; de Winter et al., 2023; Yan et al., 2020, 2021; Zhao et al., 2021), even for pre-Pleistocene 'deep time' periods. Decade-long, sub-daily-resolved records can be used to examine phenomena such as (palaeo-)ENSO, seasonal aspects of palaeoclimate as well as short-term extreme weather events (Arndt et al., 2025).

*Tridacna* are mixotrophic clams that can obtain nutrition via filter feeding and photosynthesis from the photosymbionts hosted in the soft tissue (Jantzen et al., 2008; Klumpp et al., 1992; Kunzmann, 2008). However, phototrophy is thought to be the main source of energy for the host (Klumpp and Griffiths, 1994), although the balance between phototrophy and heterotrophy might vary between species (Jantzen et al., 2008) and turbidity of the habitat (Mills et al., 2023). Overall, light availability is known to have a positive impact on calcification (Rossbach et al., 2019; Sano et al., 2012; Warter et al., 2018).

The shells of *Tridacna* contain visible patterns that provide insights into their growth and environmental history, with growth bands representing different time intervals. Banding patterns on the millimetre to centimetre scale are visible to the naked eye and may be yearly (Ayling et al., 2015; Pätzold et al., 1991; Warter et al., 2015; Welsh et al., 2011) or seasonal, with two bands per year being observed (Arndt et al., 2025; Ma et al., 2020). Bands on the micrometre scale, visible under a microscope, show daily growth (Aharon and Chappell, 1986; Arndt et al., 2023; Hori et al., 2015; Pätzold et al., 1991; Warter et al., 2018; Watanabe and Oba, 1999; Yan, 2020). All of these banding patterns within the shell are linked to changes in the shell's microstructure (Brosset et al., 2025; Mills et al., 2023, 2024).

By counting the visible daily growth bands, the age and growth rates of giant clams can be determined (Duprey et al., 2015; Fursman et al., 2025; Gannon et al., 2017; Sano et al., 2012; Zhao et al., 2023). *Tridacna* shells also exhibit daily compositional cycles, most prominently in Mg/Ca and Sr/Ca (Brosset et al., 2025; Hori et al., 2015; Sano et al., 2012; Warter et al., 2018; Warter and Müller, 2017; Yan, 2020). These elemental cycles can be detected even if daily bands are poorly visible, such as in fossil giant clams (Arndt et al., 2023). Therefore, elemental ratios can be used to help quantify growth rates and constrain the age of the specimen, e.g. using the Daydacna Python script (Arndt et al., 2023; Arndt and Coenen, 2023). However, the causes for the visible day and night banding as well as the related cycles in El/Ca ratios remain unclear.

A difference in growth rate (Sano et al., 2012; Warter et al., 2018), shell structure (Agbaje et al., 2017; Brosset et al., 2025; Mills et al., 2024) and organic content (Liu et al., 2022) have been observed between areas of the shell grown at day versus night. The varying incorporation of elements into the shell between day and night might be linked to these observations and caused by associated physiological, environmental or chemical factors. However, there is a lack of studies that unequivocally discern El/Ca changes between day and night growth in *Tridacna* under controlled experimental conditions. To the best of our knowledge, the study by Warter et al. (2018) is the only one to date that demonstrated an increase in Sr/Ca and Mg/Ca during nighttime calcification in *Tridacna*, using isotopically-labelled seawater during an unintended nighttime culturing interruption. However, the study could not rule out potential stress-related factors that might have overprinted the results.

In this study, we expand on the previous work of Warter et al. (2018) and present the results of a specifically designed culturing experiment with *Tridacna* where day and night growth periods were individually and alternatingly marked using the isotopic tracer [135]Ba. Via spatially/time-resolved elemental analysis by LA-ICPMS

at <2 μm resolution, the respective identification of day and nighttime trace elemental signatures of the shell was possible. Isotopic tracers such as [135]Ba and [87]Sr have previously been used to identify specific growth domains in foraminifera (Evans et al., 2016; Fehrenbacher et al., 2017; Hauzer et al., 2018, 2021; Levi et al., 2019) and giant clams (Warter et al., 2018), such that this approach is expected to yield new insights into calcification dynamics. With this tracer experiment, we therefore aim to provide unprecedented insights into changes in growth rate, elemental uptake from the surrounding seawater and element incorporation into *Tridacna* shells between day and nighttime.

## 2 Material and Methods

### 2.1 Culturing

Eight juvenile giant clams with lengths of 3 to 4 cm were purchased from an aquarium supplier and transported to the Hebrew University of Jerusalem, where all culturing experiments took place. The clams' species were identified by Dan Killam based on the dichotomous key of Neo (2023), with seven clams being *Tridacna maxima* and one *Tridacna squamosa* (see Tab. S1). Prior to the onset of the controlled day-night culturing experiments (see below), the clams' calcification performance was monitored over 15 days. The culture conditions (detailed below) were optimised for high calcification rates by adjusting water temperature, alkalinity, light availability and feeding during this interval.

The seawater used for the culturing of the clams was retrieved from the Gulf of Eilat. Its salinity was lowered to 37 on the practical scale by mixing 10 l of Eilat sea water (ESW), characterized by a salinity of 40.7, with 1 l of distilled water for each of four 11 l seawater reservoirs, kept in separate plastic containers. Two of the 11 l reservoirs were spiked with 0.55 ml of dissolved $^{135}BaCO_3$ (240 $^{135}BaCO_3$ μg/ml; 93.5% enriched; Oak Ridge National Laboratory, USA). In the 'spiked' seawater, the naturally minor Ba-isotope [135]Ba (6.59% natural abundance) was enriched more than tenfold. Based on a [Ba] value of ESW of 9.9 ng/g (Evans et al., 2015b), this yields a $^{135}Ba/^{138}Ba$ of approximately 1.24 in the 11 l of spiked, modified ESW. However, we stress that accurate knowledge of this ratio in the culture seawater was not required (other than being approximately tenfold higher than the natural $^{135}Ba/^{138}Ba$ of 0.092 for ease of measurement), as it solely serves to unequivocally between the respective growth increments of the cultured clams. Assuming a natural 10% inter-annual variability of [Ba] in ESW, the associated $^{135}Ba/^{138}Ba$ range of spiked, modified ESW is ~1.13 to 1.35, with Ba mass bias contributing another ~1.5% uncertainty only on this isotopic ratio, which is negligible in view of the ESW [Ba] uncertainty. Two further 11 l plastic containers with normal, non-spiked seawater were used. Night and day water was separated, hence the requirement for four different seawater reservoirs, namely 'night $^{135}Ba$-spiked water', 'night non-spiked water', 'day non-spiked water' and 'day $^{135}Ba$-spiked water'.

In order to supply nutrients to the organisms, food (Fauna Marin, type "coral sprint") was added to the reservoir water. This fine powder consists of 85% protein, 11% fat, 3% fibre and 1% ash, as well as the following additives per 1 kg: 600 i.u. vitamin D3 (E671), 50 mg iron sulphate monohydrate, 2.2 mg calcium iodate, 6 mg copper sulphate, 17 mg manganese monohydrate, 120 mg zinc monohydrate, and 57 mg antioxidants. The recommended dosage of one measurement cup every two days for 500 l of water was scaled down accordingly for the 11 l water reservoirs. Initially, after purchase, all clams grew in natural ESW in a 400 l aquarium but were in turn transferred to culturing jars. To the large aquarium filled with artificial sea water, 10 ml of "Reef Energy Plus" (Red Sea), containing carbohydrates, amino acids, fatty acids, and vitamins, are added daily during the week.

All clams underwent a 78-hour 'spiking' procedure. This included growing in the culturing jars with modified ESW enriched with $^{135}$Ba isotope tracer for 42-hours under optimized growth conditions, i.e. minimal stress, after which they were transferred back to an ESW aquarium to grow in non-spiked water for 36 hours. The optimized growth conditions included initial alkalinity above 2 mEq/kg, a temperature of 28 °C and coral food being present in the water.

During the main experimental period, all eight clams were cultured in pairs in four jars for a duration of 72 hours. The absence of other organisms in the experimental setup is relevant for growth rate analysis via alkalinity measurements, not only to exclude calcification from other organisms but also to avoid the nitrogen cycle impacting alkalinity measurements. The clam sample IDs include the jar number, the number of the clam within the jar and whether a specimen was exposed to spiked seawater during the day (DAS) or night (NIS) (Tab. 2). For example, sample "1.1 NIS" refers to the first jar, first clam that received $^{135}$Ba-spiked water at night. We note that the laboratory sample ID (see Tab. S1) is based on markings drawn on the shells for differentiation during culturing, and therefore differs from the sample ID used here. The culture jars have a water capacity of 630 ml, were sealed airtight and placed in a temperature bath. Water was pumped through the jars at a rate of 210 ml/h, resulting in a 3 h residence time. This design, in which there was a continuous inflow of reservoir water, ensured continuous nutrient availability. All jars were kept under the same conditions regarding light, temperature and salinity. The clams were exposed to light with a photon flux density of 400 µmol/(m$^2$·s) (measured on top of the jars) for 12 h, from 7:30 to 19:30. The temperature was maintained at 28 °C. The four jars were exposed to an alternating sequence of $^{135}$Ba-spiked and non-spiked water (Fig. 1); jars 1 and 2 were exposed with seawater spiked with $^{135}$Ba during the night (19:30 to 7:30), while non-spiked seawater was present in jars 3 and 4 at night. During the day (7:30 to 19:30) jars 1 and 2 were filled with non-spiked seawater, while jars 3 and 4 were exposed to seawater spiked with $^{135}$Ba.

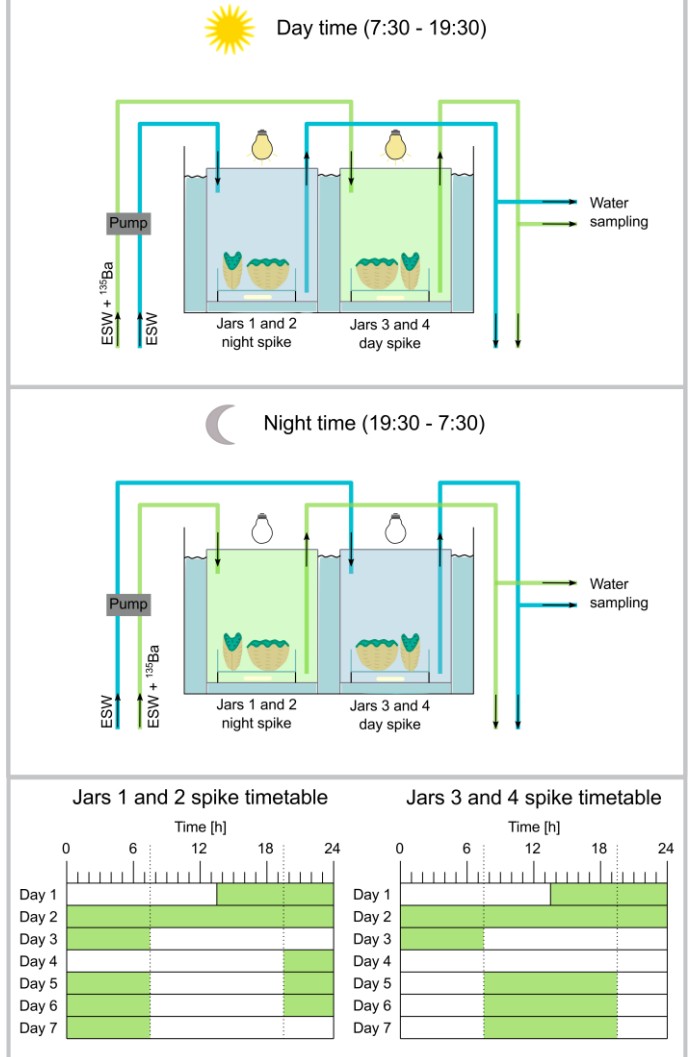

**Figure 1: Experimental setup of the culturing tanks: A: Setup during daytime, jars number 1 and 2 (represented by one jar in the sketch) contain Eilat Sea Water (ESW), while ESW spiked with the [135]Ba tracer is introduced into jars number 3 and 4 from 7:30 to 19:30. During this time, the lights over the culturing jars are on. B: During nighttime (19:30 to 7:30) the source is reversed: ESW spiked with the [135]Ba tracer is introduced into jars number 3 and 4, while jars number 1 and 2 receive non-spiked ESW and the lights remain off. A transition period where water is pumped in and out of the jars takes place in the mornings and evenings from ~7:00 to 7:30 and 19:00 to 19:30, respectively. Each jar contains two clams sitting on one petri dish. The water within the jars is stirred with a magnetic stir bar. C: Timing of the alternating seawater exposure between [135]Ba-spiked and non-spiked water (the green shaded blocks indicate [135]Ba-enriched seawater) for all four jars: After the initial spiking period of 72 hours, jars 3 and 4 received spiked water during the day from 7:30 to 19:30, as indicated by the dashed line. Jars 1 and 2 received the opposite spiked seawater treatment.**

The initial reservoir water was recycled by mixing the outflow from the jars with the remaining respective reservoir water, and aerated for 15 min. By reintegrating the outflow (~2 liters) to the respective 11 l reservoirs, nutrient concentrations within the reservoir waters were slightly diluted over the 3 day experiment, although the food within the reservoir water was sufficient throughout the culturing interval according to the aquarium food provider. The reservoir waters were sampled every morning and evening, before the new water source was connected. Final outflow water samples were taken directly from the jar outflow before the lights turned on in the morning and off in the evening.

Oxygen, pH and alkalinity measurements were performed every 12 hours on both the initial and final water samples. The resulting differences, namely Δoxygen, ΔpH and Δalkalinity, for each jar are displayed in Tab. 2.

Oxygen was monitored via an YSI ProODO meter, pH was measured using a WTW pH 340i pH meter, with measurements reported on the NIST scale, while alkalinity measurements were performed using a Metrohm 716 DMS titrino titrator. The alkalinity difference between the beginning and end value of the experiments was used to calculate the $CaCO_3$ uptake and the resulting percentage of growth per day and clam (see Tab. 2).

After the clams were sacrificed, the shells were separated from the soft tissue and brushed clean using tap water. Cleaned shells were embedded in resin and cut along the maximum growth axis. Thin sections with a thickness of 50 μm were prepared and polished using a 3 μm diamond suspension.

## 2.2 Spatially-resolved elemental analysis by LA-ICPMS

Laser-ablation inductively-coupled-plasma mass spectrometry (LA-ICPMS) was performed on thin sections using a novel custom-built dual-wavelength (157 & 193 nm) LA–ICP–MS/MS system, operated at 193 nm, coupled to an inductively coupled plasma tandem mass spectrometer, operated in single-quadrupole mode (Erhardt et al., 2025). More specifically, this comprises a modified RESOlution-SE LA-system (Applied Spectra Inc., formerly Resonetics LLC), featuring a Coherent ExciStar 500 excimer laser (instead of an Atlex ATL laser) and a Laurin Technic S-155 two-volume LA cell (Müller et al., 2009) that is linked to an Agilent 8900 ICP-MS/MS. Ablation took place in an He atmosphere (0.35 l/min) to which Ar (1.00 l/min) was added, with $N_2$ (3.5 ml/min) added downstream of the ablation cell to enhance sensitivity and plasma stability (see Tab. S2). LA-ICPMS instrument tuning used a 50 μm round spot, with 10 Hz and 3 J/cm$^2$ on NIST SRM612 to achieve tuning conditions characterized by 0.07% oxide rate (ThO$^+$/Th$^+$), 0.5% doubly-charge rate (m/z=22/44), 0.18 $^{38}$Ar/$^{80}$Ar$_2$ and 99% $^{232}$Th/$^{238}$U ratio with an $^{238}$U-signal of around 1 Mcps. The analyses were broadly based on the methodology in Warter et al. (2018) by performing laser ablation in slow continuous profiling mode with a rotatable slit.LA tracks that were set perpendicular to the daily banding, starting at the final growth segments of the outermost shell and progressing directly opposite to the direction of growth (Fig. 2). To maximize spatial resolution while maintaining suitable instrument sensitivity (crucial in view of the anticipated low Ba concentrations of these samples of around 2 μg/g), we used a rotating rectangular mask that resulted in a 1.25 × 50 μm laser spot on the sample. The ablation area is equivalent to a 9 μm round laser spot and thus provides more than sevenfold improved spatial resolution in the direction of growth, which is essential to resolve the daily growth segments ranging between ~5 to 20 μm. The rotatable slit was aligned to be parallel to the daily banding throughout the measurements (Fig. 2). Differing from Warter et al. (2018), the laser and ICPMS settings were optimized to match the 800 ms washout time of the sample introduction system (full width at 1% maximum of the single pulse response). To do so, the overall ICPMS sweep time was set to 206 ms for the following monitored isotopes (and dwell times): m/z $^{11}$B (37 ms), $^{23}$Na (16 ms), $^{24}$Mg (18 ms), $^{43}$Ca (16 ms), $^{88}$Sr (16 ms), $^{135}$Ba (47 ms) and $^{138}$Ba (42 ms). In turn, the laser beam was scanned at 1.517 μm/s to yield four sweeps per laser beam width, with the laser triggered five times during every sweep using a QuadLock device (Norris Scientifc (Norris et al., 2021)), resulting in a repetition rate of 24.27 Hz. The QuadLock aligns the laser firing rate to the ICPMS, eliminating any aliasing artifacts in the data (Müller et al., 2009; Norris et al., 2021). The on sample fluence was set to ~4.5 J/cm$^2$. Each ablation track was pre-cleaned with 100 Hz repetition rate with a spot overlap of 90%, i.e. ten shots deep, equivalent to 1.6 μm at this fluence (Coenen et al., 2024).

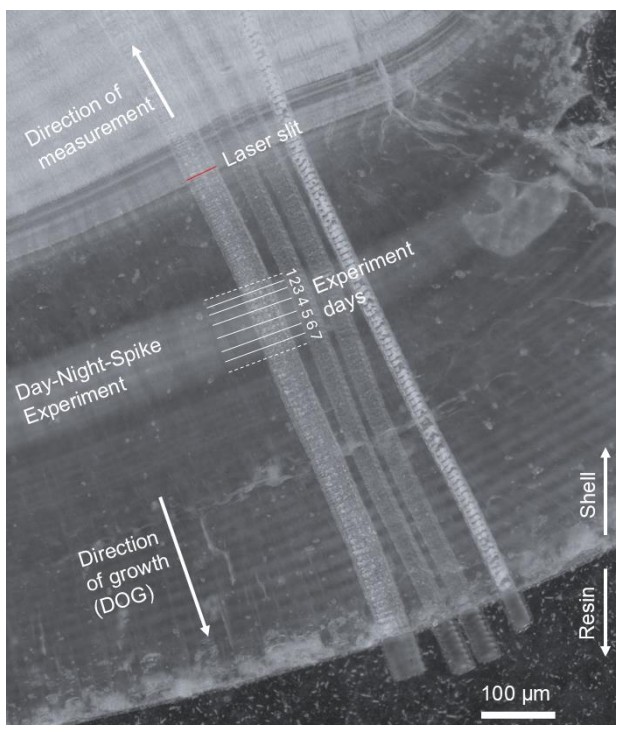

**Figure 2: Microscope image of the thin section of sample 3.1 DAS with laser-ablation paths, of which the leftmost reflects the data presented in this study. The brighter shaded domain that represents the day-night bands is indicated, including experiment days; following the experiment reported herein, the shells continued growing in a different experimental setup whose results are beyond the scope of this study and which will be reported elsewhere. The rotatable 1.25 × 50 µm laser slit, indicated by the red bar, is set parallel to the daily bands, with the laser-ablation path progressing parallel (but opposite to) the direction of growth.**

The measurements were quantified following Longerich et al. (1996) using NIST SRM612 as the external standard, with the reference values from Jochum et al. (2011), except for Mg for which the updated value of 62.4 µg/g from Evans and Müller (2018) was used. $^{43}$Ca was used as the internal standard.

Processing and visualization of LA-ICPMS data were performed utilizing the Iolite 4 software (Paton et al., 2011) and Python, with the support of libraries such as NumPy (Harris et al., 2020), Pandas (Reback et al., 2022) and Matplotlib (Caswell et al., 2022).

To evaluate data quality, i.e. accuracy and precision, measurements of the carbonate standard MACS-3 nanopellet and the MPI-DING glass standard K2L-G were performed. In the case of MACS-3, boron reference values are taken from previous LA-ICPMS analyses (Jochum et al., 2012), while the Na, Mg, Sr, Ba reference values are solution ICPMS-based data from Stephen Wilson (Jochum et al., 2012). For KL2-G we use the preferred values from the GeoReM database version 658 (Jochum et al., 2006). These standards were measured and processed in the same way as the samples. Accuracies range from -2.2 to -13.0% for MACS-3NP and -7.4 to 15.5% for KL2-G, while precision ranged between 1.5 and 10.8% 2 RSD for MACS-3NP and 1.8 and 22.2% 2 RSD for KL2-G (see Tab. 1).

**Table 1: Assessment of analytical accuracy and precision based on repeat measurements of the carbonate standard MACS-3 and the glass standard KL2-G. Reported values include measured means and associated ±2 SD, compared to published reference values (Jochum et al., 2012 (for MACS-3 B); Jochum et al., 2006 (for KL2-G); Stephen Wilson, see Jochum et al., 2012 (for MACS-3 Na, Mg, Sr, Ba)). Accuracy is expressed as the deviation from the reference value, and precision as the reproducibility (±2 RSD) of the measurements.**

| MACS-3 | Measured mean [μg/g] | ± 2 SD [μg/g] | Reference value [μg/g] | ± 2 SD [μg/g] | Accuracy [%] | Precision [% 2 RSD] |
|---|---|---|---|---|---|---|
| B | 8.63 | 0.47 | 8.90 | | -3.0 | 10.8 |
| Na | 5133 | 240 | 5900 | 800 | -13.0 | 9.2 |
| Mg | 1717 | 45 | 1756 | 272 | -2.2 | 5.2 |
| Sr | 6396 | 140 | 6760 | 700 | -5.4 | 4.5 |
| Ba | 56.3 | 0.6 | 58.7 | 4.0 | -4.1 | 2.3 |

| KL2-G | Measured mean [μg/g] | ± 2 SD [μg/g] | Reference value [μg/g] | ± 2 SD [μg/g] | Accuracy [%] | Precision [% 2 RSD] |
|---|---|---|---|---|---|---|
| B | 3.15 | 0.35 | 2.73 | 0.28 | 15.5 | 22.2 |
| Na | 16136 | 820 | 17434 | 593 | -7.4 | 10.2 |
| Mg | 42628 | 2000 | 44263 | 543 | -3.7 | 9.4 |
| Sr | 352 | 3 | 356 | 8 | -1.0 | 1.8 |
| Ba | 121.1 | 1.8 | 123.0 | 5.0 | -1.5 | 3.0 |

# 3 Results

## 3.1 Water parameters

A clear difference between night and daytime calcification is seen in the seawater carbonate chemistry and dissolved $[O_2]$ measurements performed in the evening and morning. During daytime, pH increases by ~0.14 units while oxygen concentrations rise by ~1 mg/l. At the same time, alkalinity is reduced by 80 μmol/kg on average, equivalent to a calcification rate of $0.09 \pm 0.01$ (2SD) wt% growth (weight increase) per clam for this 12 h interval. The shell weight was estimated as 60% of the measured weight of the clam with soft tissue, with the resulting

average shell weight of 5 g used to calculate calcification rates. In contrast, during the night, pH decreased by 0.17 units while oxygen concentrations are ~1.35 mg/l lower than in the morning, compared to the previous evening. The average alkalinity reduction was only 16 μmol/kg, resulting in an average calcification rate of $0.02 \pm 0.01$ (2SD) wt% per clam per 12 h. While the clams calcify both during day and night, the daytime calcification rates are almost fivefold higher.

**Table 2: Changes in water chemistry and calcification rates of the clams that were alternately exposed to $^{135}$Ba-spiked and non-spiked seawater for 12 h each during daytime and nighttime. The table indicates which clams were cultured in each jar, the spike condition (presence or absence of $^{135}$Ba), and the measured differences in pH, oxygen, and alkalinity over each 12-hour interval (note that all alkalinity anomalies were negative, thus implying calcification took place in all cases). Calcification rates were calculated from alkalinity changes and are expressed as the percentage of growth per clam per 12 hours. The average calcification of all eight clams is provided for every day and night interval.**

| | Jar | Clams | Spike | Δ pH | Δ oxygen [mg/l] | Δ Alkalinity [µMol] ± 2SD | Calcification [%/clam and 12 h] ± 2SD | Average calcification [%/clam and 12 h] ± 2SD |
|---|---|---|---|---|---|---|---|---|
| **Night 4** | J1 | 1.1 NIS, 1.2 NIS | $^{135}$Ba | -0.189 | -1.82 | 11 ± 4 | 0.013 ± 0.005 | 0.02 ± 0.01 |
| | J2 | 2.1 NIS, 2.2 NIS | $^{135}$Ba | -0.188 | -1.84 | 13 ± 4 | 0.017 ± 0.004 | |
| | J3 | 3.1 DAS, 3.2 DAS | | -0.208 | -0.73 | 20 ± 5 | 0.023 ± 0.006 | |
| | J4 | 4.1 DAS, 4.2 DAS | | -0.2055 | -1.37 | 16 ± 2 | 0.019 ± 0.003 | |
| **Day 5** | J1 | 1.1 NIS, 1.2 NIS | | 0.1325 | 0.83 | 89 ± 4 | 0.103 ± 0.004 | 0.10 ± 0.01 |
| | J2 | 2.1 NIS, 2.2 NIS | | 0.1395 | 1.18 | 86 ± 3 | 0.106 ± 0.004 | |
| | J3 | 3.1 DAS, 3.2 DAS | $^{135}$Ba | 0.1405 | 1.03 | 80 ± 2 | 0.092 ± 0.003 | |
| | J4 | 4.1 DAS, 4.2 DAS | $^{135}$Ba | 0.1785 | 1.07 | 85 ± 2 | 0.101 ± 0.002 | |
| **Night 5** | J1 | 1.1 NIS, 1.2 NIS | $^{135}$Ba | -0.1865 | -1.67 | 24 ± 4 | 0.028 ± 0.004 | 0.02 ± 0.01 |
| | J2 | 2.1 NIS, 2.2 NIS | $^{135}$Ba | -0.2035 | -1.86 | 21 ± 5 | 0.026 ± 0.006 | |
| | J3 | 3.1 DAS, 3.2 DAS | | -0.2055 | -1.38 | 16 ± 4 | 0.019 ± 0.005 | |
| | J4 | 4.1 DAS, 4.2 DAS | | -0.1755 | -1.26 | 10 ± 4 | 0.012 ± 0.005 | |
| **Day 6** | J1 | 1.1 NIS, 1.2 NIS | | 0.1885 | 0.23 | 85 ± 3 | 0.098 ± 0.004 | 0.09 ± 0.01 |
| | J2 | 2.1 NIS, 2.2 NIS | | 0.1685 | 0.31 | 70 ± 4 | 0.087 ± 0.004 | |
| | J3 | 3.1 DAS, 3.2 DAS | $^{135}$Ba | 0.1455 | 0.95 | 74 ± 6 | 0.085 ± 0.007 | |
| | J4 | 4.1 DAS, 4.2 DAS | $^{135}$Ba | 0.1465 | 1.08 | 72 ± 3 | 0.085 ± 0.003 | |
| **Night 6** | J1 | 1.1 NIS, 1.2 NIS | $^{135}$Ba | -0.1335 | -0.91 | 19 ± 3 | 0.022 ± 0.003 | 0.02 ± 0.01 |
| | J2 | 2.1 NIS, 2.2 NIS | $^{135}$Ba | -0.1105 | -0.82 | 13 ± 1 | 0.016 ± 0.001 | |
| | J3 | 3.1 DAS, 3.2 DAS | | -0.1235 | -1.31 | 21 ± 2 | 0.024 ± 0.002 | |
| | J4 | 4.1 DAS, 4.2 DAS | | -0.1215 | -1.18 | 12 ± 3 | 0.015 ± 0.003 | |
| **Day 7** | J1 | 1.1 NIS, 1.2 NIS | | 0.118 | 0.95 | 95 ± 5 | 0.110 ± 0.006 | 0.09 ± 0.01 |
| | J2 | 2.1 NIS, 2.2 NIS | | 0.121 | 1.12 | 64 ± 4 | 0.080 ± 0.005 | |
| | J3 | 3.1 DAS, 3.2 DAS | $^{135}$Ba | 0.127 | 1.31 | 79 ± 1 | 0.091 ± 0.001 | |
| | J4 | 4.1 DAS, 4.2 DAS | $^{135}$Ba | 0.0945 | 0.89 | 79 ± 3 | 0.094 ± 0.003 | |

## 3.2 Spatially-resolved compositional data (LA-ICPMS)

The elemental ratio data from the day-night spiking experiment can be found in Tab. S3 and are displayed in Fig. 3. Overall, the B/Ca values span from 0.04 mmol/mol to 0.34 mmol/mol, with an average of 0.13 mmol/mol. Na/Ca values range between 17.2 to 29.8 mmol/mol, with an average of 24.0 mmol/mol. Mg/Ca values vary nearly tenfold between 0.25 and 2.21 mmol/mol, with an average of 0.77 mmol/mol. The average Sr/Ca value is 1.57

mmol/mol, with a minimum of 0.82 and maximum at 2.4 mmol/mol. Ba/Ca values vary from 0.21 to 2.09 µmol/mol, with an average of 0.77 µmol/mol. (Fig. 3).

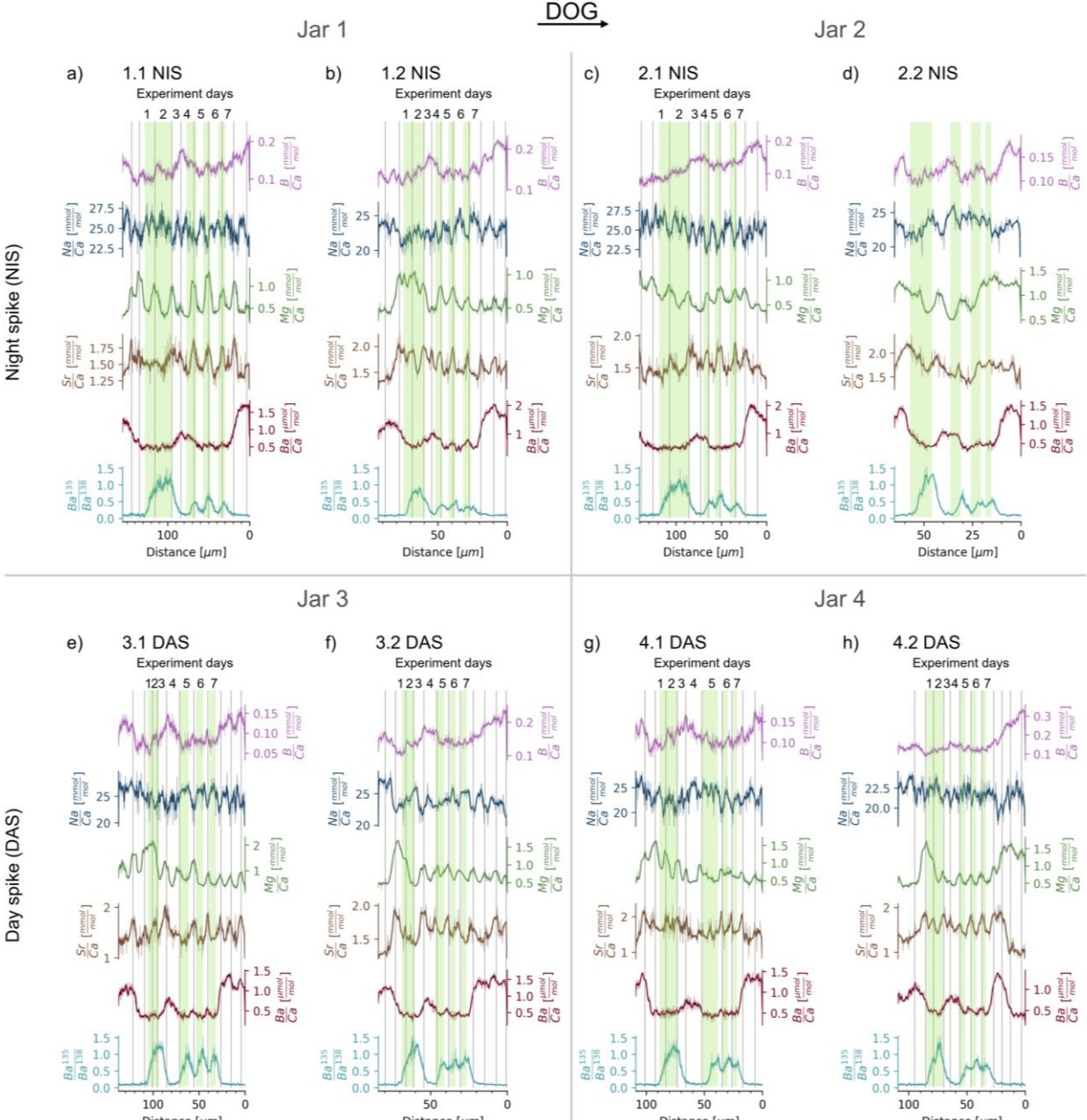

**Figure 3: Spatially-resolved El/Ca (B, Na, Mg, Sr, Ba) and $^{135}$Ba/$^{138}$Ba isotope ratios for all cultured clams.** The latter varies between the natural ratio of 0.092 and approaches that of the spiked water of ~1.24. Datasets a-d) represent the shells grown with $^{135}$Ba-spiked water during the night at experiment days 4 to 6 ('NIS'), with two jars and two clams per jar. Datasets e-h) represent the shells that were exposed to the $^{135}$Ba-spiked water during the day from days 5 to 7 ('DAS'), also with two jars and two clams per jar. Areas grown during exposure to $^{135}$Ba-spiked water are indicated by green bars. $^{135}$Ba/$^{138}$Ba intake during the 12 h interval is higher for day-spiked clams than night-spiked clams. Daily cycles are visible in Mg/Ca and Sr/Ca for 7 out of 8 clams. Clam 2.2 NIS, the only *Tridacna squamosa,* grew only about half compared to its 'partner clam' 2.1 NIS and does not display daily cycles. For the other 7 shells, vertical lines indicate Mg/Ca maxima, and if uncertain, Sr/Ca maxima, which thus delineate the daily growth increments. Note that except for $^{135}$Ba/$^{138}$Ba the x-y axis scaling is not uniform between the clams but optimized for best data visibility.

The initial introduction of the clams to $^{135}$Ba-spiked water for 42 h on experiment day 1 and 2 shows that most shells reach a $^{135}$Ba/$^{138}$Ba plateau close to the calculated $^{135}$Ba/$^{138}$Ba ratio of the water of ~1.24 after 24 h. However, shells NIS 1.2 and NIS 2.1 only reach $^{135}$Ba/$^{138}$Ba plateau values of ~0.8 and 1.0, while DAS 3.2 and DAS 4.2 reach 1.3 and 1.4, respectively. The latter might indicate that the $^{135}$Ba/$^{138}$Ba ratio of 1.24, calculated for the spiked

water, was slightly underestimated by overestimating the initial ESW [Ba]. After the last exposure to the spiked water (experiment day 7) , the $^{135}$Ba/$^{138}$Ba ratio returns to a ratio of ~0.1, i.e., approaching the natural ratio of 0.092, within 24 h in the night-spiked and within 12 h in the day-spiked clam shells.

     The data from experiment days 4 to 7 show that $^{135}$Ba is incorporated into the shells of clams subjected to the tracer both during the day (7:30 to 19:30) as well as during the night (19:30 to 7:30). The $^{135}$Ba/$^{138}$Ba ratio delineates the

relative growth between day and night, with daytime $^{135}$Ba uptake resulting in measured shell $^{135}$Ba/$^{138}$Ba ~1.7 times higher than those exposed to $^{135}$Ba at night. The cyclical $^{135}$Ba/$^{138}$Ba ratio variations are 0.2 to 0.7 and 0.4 to 1.1 for night and day-spiked clams, respectively.

     In most clams, Mg/Ca and Sr/Ca show a regular cyclicity that can be clearly identified as daily, based on the timing of the isotopic tracer introduction. Unequivocally, both elemental ratios show the highest values at the end of the

night (Fig. 4a). Daily cyclicity is less clear in Na/Ca, for which the nighttime growth seems to be characterised by domains with decreasing Na/Ca, while Na/Ca is increasing during the day. Daily cycles in B/Ca and Ba/Ca are less clear, while the ratios vary strongly between the portions of the shell grown in the aquarium and the culturing jars. The specimens grew in a large artificial seawater aquarium with other clams and corals before experiment day 1 and after experiment day 7, residing in a smaller separate aquarium with ESW during experiment days 3 to

4 and in the culturing jars with modified (reduced salinity) ESW during experiment days 1 to 2 and 5 to 7 (Fig. 1). The observed changes in B/Ca and Ba/Ca (Fig. 3) may therefore be caused by the slightly different seawater compositions in these different growth environments.

     We observe that Mg/Ca and Sr/Ca, and less clearly B/Ca and Ba/Ca, are in phase with $^{135}$Ba/$^{138}$Ba for night-spiked shells (e.g. in NIS 1.1, Fig. 4a). In contrast, day-spiked shells display an anti-phased relationship (e.g. in DAS 3.1,

Fig. 4b). Therefore, a $^{135}$Ba increase at night is seen in the shell, approximately coinciding with the minimum to maximum of both Mg/Ca and Sr/Ca cycles. In contrast, the increase in $^{135}$Ba during the day coincides with (or shortly follows) the shift from the Mg/Ca (or Sr/Ca) maximum to respective minimum (or shortly thereafter in each case).

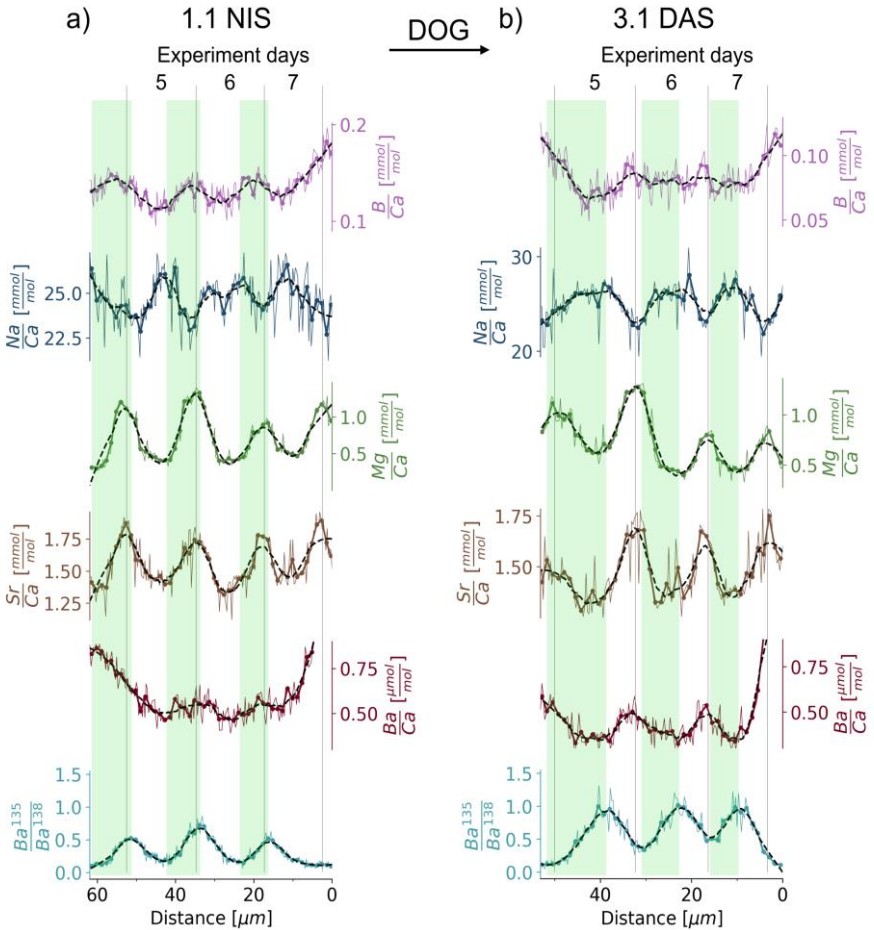

**Figure 4: Expanded view of El/Ca and [135]Ba/[138]Ba ratios for the sections of the shell grown during the day-night-spiking experiment for two representative examples, namely 1.1 NIS and 3.1 DAS. A 4-point running mean is plotted in addition to the original data. Each 4-point-mean data point is indicated by a circle, resulting in one circle every 1.25 μm which is the same as the width of the laser beam. The dashed line represents a quadratic polynomial fit to the 4-point mean signal using a Savitzky-Golay filter (Savitzky and Golay, 1964), which smooths the data while preserving important features of the signal. Areas grown during exposure to [135]Ba-spiked water are indicated by green bars.**

## 4 Discussion

### 4.1 Reproducibility

All eight clams grew in the same environmental conditions in terms of temperature, light and food availability, and only the timing of the introduction of [135]Ba-spiked water varied systematically. Four clams in two jars received a [135]Ba-spike at night and daytime, respectively. With two clams per jar and two jar per treatment we can compare whether the elemental ratio patterns are the same between clams grown in the same jar and in different jars with the same treatment. Furthermore, this provides a replication of the water chemistry and estimated growth rates, which, in our experimental setup, reflect the mean of the two clams inhabiting the same jar. Some patterns consistently emerge between the two sets of four replicates, such as the overall incorporation of the isotopic tracer, daily elemental cycles and overall changes in shell chemistry between aquarium and culturing water (Fig. 3). On a more detailed level, there are clear differences between the individual clams regarding overall growth, growth pattern and element incorporation even between pairs kept within the same jar (see 1.1 vs 1.2 NIS, 2.1 vs 2.2 NIS, 3.1 vs. 3.2 DAS). During the seven days of experiment, the clams' shells grew between 55 to 115 μm in the direction of growth. Similarly, daily growth rates range from 6 to 21 μm/day and vary by up to 9 μm/day even

within one clam (Fig. 3e). As environmental factors were identical for the two clams in one jar, such variability is presumably related to the individual physiology of the clams.

The culturing conditions of this experiment were based on the findings of Warter et al. (2018) regarding temperature, light and food availability, which were optimized for high calcification rates through several experiments. The resulting calcification patterns may therefore not be uniformly applicable to all species, sizes and temperature regimes, although the parameters utilised here are within the range of those that *Tridacna* experience in their natural environment. In addition, we note that the daily cycle amplitudes and absolute El/Ca values from the cultured *Tridacna* of this study overlap well with data from naturally grown samples, namely those from the late Miocene *Tridacna* of Arndt et al. (2025) as well as the recent and fossil clams from Warter and Müller (2017), which were all measured using a similar analytical approach. However, we do observe that the Ba/Ca minima in all clams are below the average minus 2SD of the values reported in the literature. Three clams (3.1 DAS, 3.2 DAS, 4.2 DAS) also exhibit unusually high Mg/Ca maxima with values 52, 20 and 17 % above the average of the literature values, respectively. Two clams (4.1 DAS, 4.2 DAS) are characterised by very low Sr/Ca minima, 34 and 32 % below the literature average (see Tab. S4). Nonetheless, the overall comparability of the data from cultured, recent and fossil clams grown in different environmental settings, combined with the variability observed in El/Ca ratios among clams grown under identical conditions in this study, underscores the significant biological influence on the El/Ca composition of clam shells, as well as the applicability of laboratory culture results to naturally grown samples.

**4.2 Analytical resolvability**

Given a sweep time of 0.206 s and a scan speed of 1.517 μm/s, the resulting sampling frequency is 0.313 μm. With a slit width of 1.25 μm this results in four sampling points per slit with. If we use the 4-point mean values, i.e. one data point per slit width we can safely detect cycles with a wavelength of greater than 3 μm. As daily cycle wavelengths range from 6 to 20 μm, all compositional cycles should analytically be well resolved.

The full signal rise takes 0.2 s (i.e., 0.3 μm) in the sampling direction and signal washout to 10% and 1% takes place in 0.5 s and 1.2 s (0.8 and 1.8 μm in sampling distance), respectively. Therefore, data points in 1.25 μm steps (~0.8 s measurement time) are not completely independent but can contain ~ 5% of the signal from the previous data point. As the resulting measured signal consist of around 95% of the actual signal at that distance, we do not expect washout to lead to substantial signal alteration.

Using a narrow but wide laser slit (1.25 × 50 μm) helps to obtain suitably high elemental signals for low [Ba] and [B] while maintaining very high spatial resolution (<2 μm). With a narrow, long slit optical alignment to daily banding is facilitated as misalignment is easy to see, however slight variations from the alignment might result in minor signal mixing between growth layers. In some shell areas the daily band structures are either slightly curved or not visible at all (Fig. 2). We aimed to position the laser slit as parallel as possible to the daily banding, avoiding these structures. Quantitatively, if the slit is 2° misaligned, it would cover ~2 μm in direction of growth, while 5° result in almost 5 μm, i.e., almost four times the laser slit width. We evaluated the microscope pictures of the laser ablation paths on the shells. As the day-night-experiment is well visible through a lighter shading in the shell (likely due to higher temperatures and faster growth rates compared to the aquarium; Fig. 2) an accurate alignment was possible. We therefore demonstrate that we measured parallel to the daily banding with occasional offset between laser slit and growth banding direction of below 2°. While limited analytical signal smoothing may have

occurred in some areas, the associated signal smoothing would be on the order of 2 μm or less, which is still below half the minimum daily growth width.

Nevertheless, the dataset contains areas with unclear El/Ca signals on scales of tens of μm, e.g. in Mg/Ca days 1 to 2 of DAS 4.1, Sr/Ca days 1 to 2 of NIS 1.1 and in Mg/Ca throughout NIS 2.2 (Fig. 3). Interestingly, at the same distance, the cycles in the other respective ratios (Sr/Ca, Mg/Ca and $^{135}$Ba/$^{138}$Ba) are resolvable. This provides additional evidence against substantial analytical smoothing of the signal as the reason for these unclear cycles. We therefore argue that the unclear cyclic patterns are not analytically caused but reflect the clams' individual growth performance and elemental uptake into the shell.

**4.3 EPF reaction and replenishment time**

The cyclic behaviour of the $^{135}$Ba/$^{138}$Ba ratio within the shells contrasts with that of the rapid switch between the endmembers of spiked ($^{135}$Ba/$^{138}$Ba = 1.24) and non-spiked ($^{135}$Ba/$^{138}$Ba = 0.092) water (Fig. 5). In most shells, it takes about one day (i.e. ~5 to 20 μm of growth depending on the specimen) until the Ba isotopic ratio of the spiked water is reached. This smoothing is, as discussed above, unlikely to be caused by limited analytical smoothing because it is similar in all clams and larger than the worst case of analytically-induced smoothing of ~3 μm. We thus deduce that the $^{135}$Ba/$^{138}$Ba ratio behaviour reflects predominantly the uptake patterns of elements into the EPF.

During the day-night-spiking, i.e. experiment days 4 to 7, the uptake of the $^{135}$Ba-isotopic tracer starts shortly after the Mg/Ca maximum (night spike) or minimum (day spike) is reached after which $^{135}$Ba/$^{138}$Ba continues to rise until the Mg/Ca minimum (night spike) or maximum (day spike). We therefore infer that during the 12 h culturing period, in the presence of the $^{135}$Ba spike, the concentration of $^{135}$Ba continuously increases in the EPF, such that the $^{135}$Ba/$^{138}$Ba value of the shell continuously rises. Similarly, washout begins rapidly, as indicated by a reduction of $^{135}$Ba/$^{138}$Ba right after the Mg/Ca minimum (night spike) or maximum (day spike) is reached. Taken together, we can unambiguously determine that the uptake of the spike occurs more quickly during daytime. Samples spiked during the day are broadly characterised by 1.7 times higher $^{135}$Ba/$^{138}$Ba ratios, which do not return to the baseline (i.e. natural) $^{135}$Ba/$^{138}$Ba at night. This indicates that the residence time of Ba in the EPF is substantially shorter during the day than at night. Assuming barium and calcium behave similarly, this aligns with our alkalinity measurements described above, which constrain a fivefold increase in calcification rate.

Despite a quick EPF 'reaction' time, indicated by rapid initial increase or decrease of the tracer within the shell, the time needed to reach the maximum $^{135}$Ba/$^{138}$Ba value is approximately one day, as also indicated by the initial 42 h tracer exposure (Fig. 5). This further supports the idea of continuous exchange between the ambient seawater and EPF and, for these small specimens, indicates a barium (and calcium) residence time in the EPF of ~24 hours.

*Tridacninae* have been observed to partially close during the night, likely as a defence mechanism against predators (Killam et al., 2023). A partial closure could affect the overall EPF volume, potentially resulting in a reduced volume present at night and thus shortening the residence time of elements in the EPF. However, data from a controlled environment without predators shows no clear diurnal change in valve opening (Rossbach et al., 2020). Nonetheless, we explore whether a diurnal change in EPF volume could explain our observations, for the theoretical case that the pumping of ions into the EPF would be equal between day and night. In this scenario, a smaller extrapallial volume should show a faster increase of $^{135}$Ba relative to $^{138}$Ba within the EPF, such that the $^{135}$Ba/$^{138}$Ba ratio in the shell should increase faster at night. Conversely, a smaller EPF volume should lead to faster washout/flushing of the EFP during the night, resulting in a faster decrease of $^{135}$Ba/$^{138}$Ba within the shell. This is

contrary to the pattern we observe, such that possible diurnal changes in EPF volume cannot be the main driver of the observed diurnal $^{135}Ba/^{138}Ba$ variability. However, this phenomenon could act to dampen the observed signal that we assign to diurnal changes in active ion transport into the EPF. Future work could test whether the barium residence time is longer for larger clams with a larger extrapallial volume, or whether the residence time is maintained as the result of a more efficient replenishment of the more voluminous EPF.

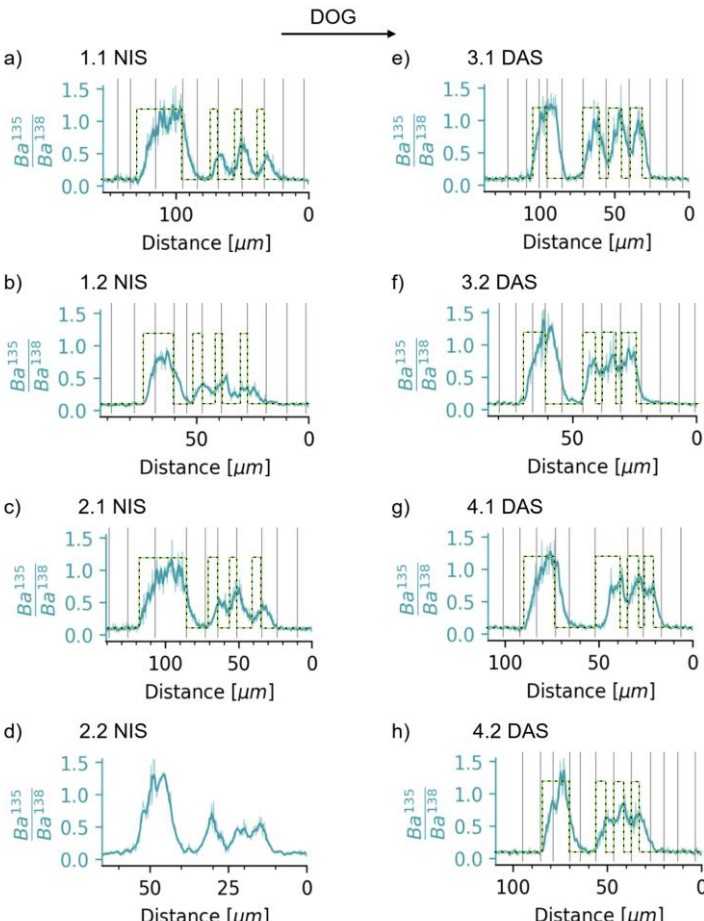

Figure 5: Measured $^{135}Ba/^{138}Ba$ ratios of all clams compared with the phases of $^{135}Ba$-spiked water introduction, marked as a green-dashed rectangular function. The two endmembers show the natural $^{135}Ba/^{138}Ba$ ratio of 0.092 and the calculated $^{135}Ba/^{138}Ba$ of the spike water of 1.24. The deviation of the measured $^{135}Ba/^{138}Ba$ ratio (blue) mostly reflects the uptake behaviour of Ba and other elements from the surrounding seawater into the EPF, with possible maximum analytical smoothing of 2-3 μm.

### 4.4 Causes for daily El/Ca cycles

In the following sections we discuss which of the possible environmental, biochemical and physiochemical factors have the clearest impact on daily El/Ca ratios. While these factors are inherently interconnected and often exhibit complex interactions, they will be discussed separately to facilitate a clearer and more detailed understanding of their potential contributions.

### 4.4.1 Environmental factors

In natural environments, co-varying parameter patterns can occur, e.g. temperature and light both positively affect growth rate in *tridacnines* up to a stress limit (Killam et al., 2021; Schwartzmann et al., 2011; Van Wynsberge et al., 2017; Warter et al., 2018). Besides temperature and light, a further environmental factor that can impact daily

elemental cycles is the availability of food that can be acquired through filter feeding. In reef environments, dissolved oxygen concentrations, which broadly reflect net productivity and thus food availability, fluctuate diurnally. Concentrations are lowest at night, rising due to photosynthesis after sunrise, peaking in the afternoon, and then declining due to respiration and oxygen degassing (Silverman et al., 2007). Given that *Tridacna* have photosymbionts and generally rely heavily on phototrophy, albeit with some species and environment-dependent variability (Jantzen et al., 2008; Klumpp and Griffiths, 1994; Mills et al., 2023), light may be a more important environmental driver of shell chemistry. It has been proposed that *Tridacna* rely more on photosynthesis for nutrition during the day and more on filter feeding during the night (Killam et al., 2023). However, giant clams grown in culture experiments in which temperature and nutrient concentrations were not varied between day and night still show daily cycles (Warter et al., 2018; Figs. 3,4). In our setup, two clams grew in one jar with a continuous throughflow of water (residence time 3 h). Thus, we assume that the presence of only two clams and the constant refreshening of the water around the clams strongly reduced the daily nutrient cycle caused by photosynthesis and respiration, compared to a reef environment. We therefore suggest that neither temperature nor nutrient availability are the key driver of heterogeneity in shell chemistry in *Tridacna*. Nevertheless, measurements of daily variability in culture water nutrient composition, which was not performed here, may provide further information on the metabolic activity of the clams and their photosymbionts. Future experiments could investigate this factor explicitly as well as its impact on shell mineralization and chemical composition, ideally in fully closed off culturing jars or jars with reduced water throughflow, containing several giant clams.

Given a diurnal change in nutrient availability is not likely to drive the observed geochemical heterogeneity, light availability is a more likely explanation for the daily variability in shell growth and element incorporation, especially given that light enhanced calcification has been observed in *Tridacna* (e.g., Ip et al., 2015; Rossbach et al., 2019; Sano et al., 2012). Recent biochemical studies investigated elemental transport through the mantle tissue within the context of the light dependent expression of channelling proteins relevant for shell formation. Indeed, multiple light enhanced calcium and bicarbonate transport mechanisms that could result in this phenomenon are active in shell formation, including voltage-gated calcium channels, $Na^+/Ca^{2+}$ exchangers, plasma membrane $Ca^{2+}$-ATPase, and bicarbonate transporters (Boo et al., 2021, 2022; Cao-Pham et al., 2019; Chan et al., 2021; Chew et al., 2019; Ip et al., 2015, 2017).

**4.4.2 Biochemical factors**

Organic material present at the calcification site or involved in 'templating' the growing shell, might be important in driving shell chemistry. During daytime, calcification is faster such that less templating organic material might be incorporated into the mineral component of tridacnid shells, while thinner bands, presumably grown at nighttime, are characterized by comparatively higher templating organic content (Liu et al., 2022). However, more dissolved organic matter may conversely be incorporated during faster growth. In the case of inorganic calcite, the addition of certain organic molecules (peptides) has been shown to increase both growth rate and Mg incorporation by reducing the dehydration enthalpy of $Mg^{2+}$ to a greater degree than $Ca^{2+}$, resulting in increased Mg uptake (Stephenson et al., 2008). As organic material in *Tridacna* is present to a greater degree in the lower growth, nighttime bands with higher Mg/Ca, we assume that the availability of organic material is not a limiting factor in determining shell growth rate on a daily basis. Irrespective, the incorporation of organic material could conceivably be important in controlling shell chemistry (to a degree) given that the concentration of key elements, typically considered to primarily substitute for $Ca^{2+}$ in the mineral lattice, may also be present in organics at relatively high

concentration. For example, seasonal growth bands in *Artica islandica* and *Acesta excavata* are associated with increased organic contents and $Mg^{2+}$ bound to the organic material, leading to increased Mg/Ca ratios during LA-ICPMS measurements (Schleinkofer et al., 2021; Schöne et al., 2010). It has also been suggested that seasonal Sr/Ca variability in *A. islandica* shells is controlled by the organic matrices at the calcification site (Shirai et al., 2014). However, given the absence of the (trace) elemental composition of tridacnid organic material, we cannot assess the degree to which this could drive daily chemical banding.

Additionally, it has been shown that amorphous calcium carbonate (ACC) acts as a precursor phase in the biomineralisation of at least some bivalves (Addadi et al., 2006; Weiss et al., 2002). However, while organic material is incorporated into ACC, the small amount of existing research has shown opposing impacts on $Mg^{2+}$ incorporation. Wang et al. (2009) found an increase in ACC Mg/Ca when organics are added, while Evans et al. (2020) found that adding amino acids reduced Mg/Ca. The latter observation can be explained by the reduction in the solution Mg/Ca activity ratio via a preferential binding of $Mg^{2+}$ with amino acids in solution, resulting in less $Mg^{2+}$ being incorporated (Evans at al. 2020). Irrespective, it is agreed that $Mg^{2+}$ binds with amino acids to ligand-ion complexes to a differential degree to $Ca^{2+}$, such that this phenomenon has the potential to impact shell chemistry in organisms that utilise an amorphous precursor. Determining the direction in which this effect may act would require assessing the competing effects of co-incorporation of metals with organics into ACC and during crystallisation versus the impact that organics have on the $Mg^{2+}/Ca^{2+}$ solution activity ratio (and therefore ACC and aragonite Mg/Ca), which is beyond the scope of this study. Studies on tridacnid shell growth have shown that the shell ultrastructure varies between day and night growth (Agbaje et al., 2017; Brosset et al., 2025; Mills et al., 2024). These shell structures are associated with organic matrix components (Kobayashi and Samata, 2006) and have been interpreted to be caused by the variability in growth rate and possibly also organic content (Mills et al., 2024).

### 4.4.3 Physicochemical factors

A diurnal change in the physiological performance of the clams, regarding the replenishment of the EPF, indicates the possibility that elemental cycles are caused by non-uniform depletion of the EPF. This Rayleigh fractionation-type behaviour (Elderfield et al., 1996; Evans et al., 2018; Ram & Erez 2025) would potentially be visible if the rate of supply versus consumption of ions to the calcification site differed during the night compared to the day. In the case that Rayleigh distillation was a key control of shell trace element chemistry, we would expect elements with partition coefficients ($K_d$ = El/Ca$_{aragonite}$ / El/Ca$_{sw}$) below 1 to be characterised by antiphase cyclicity relative to those with $K_d$ above 1. However, day-night cycles in Mg/Ca are in phase with those in Sr/Ca and Ba/Ca and opposed to those in Na/Ca, even though Na and Mg have partition coefficients well below 1, while Sr is slightly above 1 and Ba well above 1 for inorganic precipitation at 25 °C (Gaetani and Cohen, 2006) and in corals (Giri et al., 2018; Ram and Erez, 2021). The pattern of Mg/Ca varying in phase with Sr/Ca and Ba/Ca indicates that a differential degree of Rayleigh fractionation at night is not the main cause of the observed day-night cycles.

In addition, we can demonstrate that the EPF is replenished by isotopically-labelled seawater during both day and night and is thus not a closed system at night. However, the differential degree of tracer uptake, with 1.7 times higher $^{135}Ba/^{138}Ba$ peaks (i.e. closer to the ratio of the labelled seawater of 1.24) in day-spiked shells compared to those spiked at night, could indicate increased seawater uptake and/or ion transport during the day (see Fig. 5).

The observed elemental banding patterns could also be caused by kinetic effects through trace element partitioning into aragonite, driven by changes in the aragonite precipitation rate. Increased precipitation rate causes increased

incorporation of $Na^+$ (Brazier et al., 2024) and $Mg^{2+}$ (Mavromatis et al., 2022), while $Sr^{2+}$ incorporation broadly decreases at higher growth rates (Brazier et al., 2023), although other studies have suggested a more complex response depending on temperature and solution chemistry (AlKhatib and Eisenhauer, 2017). As seen in a previous study (Arndt et al., 2023), Na/Ca in *Tridacna* appears to increase while Mg/Ca and Sr/Ca decrease with increased calcification rates. While uncertainties remain regarding kinetically-driven element incorporation, Na/Ca and Sr/Ca therefore appear to follow the suggested trends, while Mg/Ca behaves inversely to the expected pattern if the banding was driven by kinetics alone. In addition, however, inorganic calcite precipitation experiments have shown that higher $Mg^{2+}$ concentrations in solution and/or in the growing solid phase results in lower growth rates, while $Sr^{2+}$ incorporation increases growth (Knight et al., 2023). Therefore, a higher $[Mg^{2+}]$ in the EPF during nighttime could be the cause rather than the effect of low growth rates, and may reconcile the antiphase relationship between Mg/Ca and Na/Ca.

Finally, increased active $Ca^{2+}$ pumping (possibly light enhanced) into the EPF during daytime would act to reduce the relative concentration of the other elements in the EPF (Sr, Ba, Mg, B) through dilution. This could explain why most elemental ratios (except Na/Ca) - independent of their fractionation factors and inorganic growth rate dynamics - decrease during the day when growth rates increase.

**4.4.4 Why is Na different?**

Na/Ca behaves differently than the other El/Ca as it decreases during the night. This may be because the $Na^+$-proton and $Na^+$-bicarbonate transporters are involved in controlling the carbonate chemistry of the EPF and maintaining charge balance in channelling $Ca^{2+}$ (Ip and Chew, 2021). In *Tridacna*, the pH of the EPF appears to be dependent, at least to a degree, on light, with pH being controlled through light dependent $NH_4^+$ channelling (Ip et al., 2006). In addition, the $Na^+/H^+$ Exchanger (βNHE) is active in pumping out $H^+$ and regulating pH within the seawater-facing epithelium of the mantle (Cao-Pham et al., 2019a), while light-dependent $Na^+/H^+$ exchangers are thought to be important in bringing inorganic carbon into the clam and to the photosymbionts (Hiong et al., 2017; Ip and Chew, 2021). $Na^+$ is also used for metal transport at the EPF-facing epithelium. Specifically, light enhanced activity of the NCX3 enzyme pumping $Ca^{2+}$ into the EPF in exchange for $Na^+$ has been shown (Boo et al., 2019; Ip and Chew, 2021). NCX3 requires the support of $Na^+/K^+$-ATPase, which transports $Na^+$ back into the EPF and also shows light-enhanced activity (Boo et al., 2017). In addition, bicarbonate transport to the EPF may be an important carbon source for calcification. This may be achieved with the electrogenic $Na^+/HCO3^-$ co-transporter, further increasing $Na^+$ in the EPF during times of increased calcification (Ip and Chew, 2021). All of these processes may lead to a higher $Na^+$ concentration in the EPF during light exposure, in addition to the kinetic effect described above.

**4.4.5 Potential mechanisms explaining elemental banding patterns**

Compared to the composition of ESW (Evans et al., 2015a; Steiner et al., 2025) Mg/Ca, Sr/Ca and Ba/Ca are strongly reduced in the shell. Specifically, the observed distribution coefficients are 0.14 for Mg/Ca, 0.19 for Sr/Ca and 0.09 for Ba/Ca (see Tab. S5). This pattern is likely caused by discrimination during ion transport into the EPF. However, we stress that we did not measure the EPF directly, such that this interpretation is based on inference from shell chemistry. With this caveat in mind, light enhanced $Na^+$ pumping coupled to $Ca^{2+}$ and $HCO_3^-$ transport might overall explain why Na/Ca is elevated during the day unlike other El/Ca (El = B, Mg, Sr, Ba). Together with a (possibly more minor) role of growth rate-related kinetic effects, we propose that light dependent $Ca^{2+}$ and $Na^+$

transport and the dilution of other (trace) elements in the EPF can explain the daily cycles in El/Ca seen here (Figs. 3, 4) and previously reported (Arndt et al., 2023; Brosset et al., 2025; Hori et al., 2015; Sano et al., 2012; Warter et al., 2018; Warter and Müller, 2017; Yan, 2020).

We assume that the same mechanism, namely the intensity of active $Ca^{2+}$ and $HCO_3^-$ transport into the EPF, directs day and night changes in El composition, growth rate, shell structure and possibly organic content. An increase in calcification rate during the day, namely fivefold higher at day than at night for the clams cultured in this study (Tab. 2), could affect the crystal structure and might lead to a relatively lower content of template organics in the shell.

Light enhanced activation of important channelling enzymes could indicate that the formation of daily elemental cycles and increments is light dependent. It has, however, also been suggested that, independent from environmental factors, circadian rhythms play an important role in dictating diurnal changes (Liu et al., 2024; Warter et al., 2018; de Winter et al., 2023), with de Winter at al. (2023) observing that a shell grown under shades does not exhibit significantly different daily cycles. Even under shaded conditions, increased photosymbiont activity during daytime could provide excess energy that can be used for active $Ca^{2+}$ and $HCO_3^-$ transport. The observed increase in oxygen concentration during daytime indicates that in our culturing setting, more oxygen is produced by the photosymbionts than is consumed by the giant clams during the day. Energy consumption and light availability seem to be strongly coupled in *Tridacna* and thus it remains difficult to distinguish the impact of metabolic activity throughout the day from that of light availability.

**4.5 Relative phasing of the daily increments and elemental cycles**

An increase in trace element concentrations in bands grown at night has previously been predicted on the assumption that thinner bands likely represent the night growth, while thicker bands represent day growth in *Tridacna* (Brosset et al., 2025; Mills et al., 2024; Sano et al., 2012). While daytime shell growth was previously assumed to be about three times higher than at night (Brosset et al., 2025), our alkalinity measurements (conducted every 12 h) allow us to determine that the eight specimens utilised here grew on average five times faster regarding calcification volume (Tab. 2). The accumulation of new shell material in the direction of growth is, however, only roughly 1.2 times higher during daytime versus nighttime (Figs. 3, 4). This could indicate that the day versus night extension of the shells area perpendicular to the direction of growth (adding to shell width) is even higher than extension in direction of growth (adding to shell thickness). If the fivefold increased shell growth at daytime would be uniform in all directions, we would expect the respective extension in direction of growth to be 1.7 (the cube root of five). As shell extension in direction of growth during daytime is only increased 1.2-fold, the remaining material must be precipitated perpendicular to the direction of growth to make the shell wider (as compared to thicker). If the shell width increases more at daytime than at nighttime, we would expect an incremental growth pattern on the micrometre scale at the outer surface of the shell. While this incremental growth pattern is poorly preserved in the thin sections (likely due to polishing), it is clearly seen as a grooved structure on photographs of the shell (see Fig. S1).

By comparing the differential incorporation of the $^{135}Ba$ isotope tracer, supplied only during day or during night, to the Mg/Ca and Sr/Ca cycles within the shells, we gain new insights about the detailed timing of the Mg/Ca and Sr/Ca increase and decrease over the course of a day. Based on this, we can unequivocally show that Mg/Ca and Sr/Ca are increasing during nighttime in the narrower low-growth bands, but Mg/Ca and Sr/Ca values are not necessarily higher at nighttime, as previously suggested (Brosset et al., 2025; Hori et al., 2015; Warter et al., 2018).

The clams that grew with the tracer introduced at nighttime (from 19:30 to 7:30) begin to incorporate the tracer into the shell approximately during the Mg/Ca and Sr/Ca minimum, with the tracer concentration decreasing within the shell shortly after the Mg/Ca and Sr/Ca maximum is reached. Similarly, the clams that grew in the presence of the tracer during the day (from 7:30 to 19:30) are characterised by a tracer onset in the shell approximately coincident with the Mg/Ca and Sr/Ca maximum. The tracer signal decreases shortly after the Mg/Ca and Sr/Ca minimum is reached. The isotopic tracer administered for 12 h each can only be incorporated as observed if the Mg/Ca and Sr/Ca maxima reflect the early morning hours or the end of the night, while the Mg/Ca and Sr/Ca minima are reached in the evening hours or towards the end of the day.

**4.6 Relevance for palaeoclimate applications**

Understanding the causes of daily element banding holds relevance for palaeoclimate applications. The daily El/Ca patterns, likely caused by metabolic activity and ion pumping, reflect seasonal patterns that have previously been investigated in fossil specimens, which also exhibit a dark-light shell banding pattern in both day-night bands and seasonal sunny-rainy-season banding (Arndt et al., 2025). While seasonal variability is certainly influenced by various factors on scales not present in daily cycles, insights into the effects of light availability and metabolic activity provide useful additional information to consider alongside environmental factors when interpreting palaeo-seasonality. Together with the analysis of recent clams grown in a natural environment (e.g., Arias-Ruiz et al., 2017; Elliot et al., 2009; de Winter et al., 2023; Yan et al., 2020), information from such short-term culturing experiments helps distinguish – or even link – biological effects and environmental signals, aiding in the production of accurate reconstructions of past conditions. The results presented here facilitate this by providing unambiguous evidence of the portion of the shell that is grown during the day.

Furthermore, daily cyclicity provides the possibility of quantifying daily cycle wavelengths as a useful tool for obtaining a high-resolution age model for geochemical datasets, independent of daily band visibility (Arndt et al., 2023). Having daily resolved palaeoclimate data at this time scale facilitates the identification of cyclic variability, including seasonality and multi-annual atmospheric and oceanographic oscillations such as the El Niño–Southern Oscillation, via spectral analysis (Arndt et al., 2025). In addition, knowledge of how giant clams build their shells daily may facilitate the interpretation of short-term geochemical signals linked to rapid changes in the environment, such as extreme weather events. All of these approaches require the driving factors behind element cyclicity to be understood within the context of the biomineralisation process of the organism. This is only possible when physiological versus kinetic chemical signals can be disentangled, which requires knowledge of how and when different domains within the shell were formed.

**5 Conclusion**

Here, we present calcification and geochemical data from eight *Tridacna* specimens grown under controlled conditions in the laboratory in seawater labelled with a [135]Ba isotope spike, introduced during either the day or night only. Our findings indicate that calcification rates in *Tridacna* are nearly five times higher during the day compared to those at night. The EPF is replenished by ions from the surrounding seawater during both day and night, resulting in daily elemental heterogeneity of the shell. We demonstrate that the uptake and washout of elements start quickly, likely within tens of minutes, but requires approximately one day to fully replace the barium pool, and assuming similar behaviour, that of calcium in the EPFs. During the daytime, B/Ca, Mg/Ca, Sr/Ca and

Ba/Ca decrease while Na/Ca increases, and vice versa during nighttime. Our data also demonstrate that the night peak, often best seen in Mg/Ca, happens in the early morning hours rather than at midnight, while the day peak reflects the evening hours rather than the middle of the day. Daily El/Ca cycles might be affected by growth rate dynamics, while the elemental composition of the EPF could in turn affect growth rates. It is, however, likely that daily El/Ca cycles are primarily caused by changes in the intensity of active $Ca^{2+}$ and $HCO_3^-$ transport into the EPF. This reduces most El/Ca-ratios by increasing $[Ca^{2+}]$ relative to the other elements during the day, with the exception of $Na^+$, which is strongly involved in $Ca^{2+}$, $HCO_3^-$ and $H^+$ transport (e.g. via $Na^+$-$HCO_3^-$ co-transport). The changes in active pumping into and from the EPF could in turn be dependent on light availability, circadian rhythms or simply an expression of the energy available for transport. We therefore propose that active pumping of $Ca^{2+}$ and $HCO_3^-$ into the EPF might also cause diurnal changes in growth rate, organic content and shell structure. Overall, this study enhances our knowledge base for utilizing *Tridacna* as a high-resolution palaeoclimate archive, particularly for sub-daily time scale applications, by improving the spatial resolution of LA-ICPMS and deepening our understanding of diurnal growth patterns, variability in elemental composition, and therefore rapid responses to environmental changes.

**Competing interests**

The authors declare that they have no conflict of interest.

**Data availability**

All data, supplementary tables and supplementary figures presented in this study (Fig. S1, Tab. S1 – S5) are available at Zenodo with licence CC BY 4.0 (*link will be inserted at later stage*).

**Authors contributions**

IA: Conceptualization, Data curation, Formal analysis, Funding acquisition, Investigation, Validation, Visualization, Writing (original draft preparation), Writing (review and editing); JE: Conceptualization, Funding acquisition, Investigation, Methodology, Project administration, Resources, Supervision, Validation, Writing (review and editing); DE: Conceptualization, Data curation, Supervision, Validation, Writing (review and editing); TE: Data curation, Investigation, Methodology, Supervision, Validation, Writing (review and editing); AL: Investigation, Validation, Writing (review and editing); WM: Conceptualization, Data curation, Funding acquisition, Methodology, Project administration, Resources, Supervision, Validation, Writing (review and editing)

**Acknowledgements**

We would like to acknowledge the Deutsche Forschungsgemeinschaft (DFG MU 3739/6-1) for their financial support of the corresponding author (IA) as well as the culturing experiments at the Hebrew University of Jerusalem (WM, JE). We also appreciate the support from the VeWA consortium (Past Warm Periods as Natural Analogues of our high-$CO_2$ Climate Future), funded by the LOEWE programme of the Hessen Ministry of Higher Education, Research and the Arts, Germany (DE, WM). We gratefully acknowledge DFG Major Equipment Grant

INST 161/1073-1 FUGG and Goethe University startup funding (both to WM). We thank Maria Bladt and Niels Prawitz for their valuable work in sample preparation and Wolfgang Schiller for photographing the shells. We highly appreciate the work done by Dan Killam to identify the *Tridacna* species. Additionally, we acknowledge Linda Marko and Alexander Schmidt for their technical support in the FIERCE laboratories. FIERCE is financially supported by the Wilhelm and Else Heraeus Foundation and by the Deutsche Forschungsgemeinschaft (DFG: INST 161/921-1 FUGG, INST 161/923-1 FUGG and INST 161/1073-1 FUGG), which is gratefully acknowledged.

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
