# Peer review of "Culturing experiments reveal mechanisms of daily trace element incorporation into *Tridacna* shells"

_EGUsphere, 2025_

## Author Response (AR1)

Dear Dr. de Winter,

Thank you for the evaluation of the manuscript and the overall positive comments. In the following we provide detailed answers to the suggestions for improvement provided by you and the reviewers (indicated in italic fond). We indicate the changes made to the revised manuscript and reference the line numbers (L).

***Editors Comments:***

*1. - Discuss the potential offset in the chemistry of the extrapallial fluid of tridacnids from seawater*

Because we have not measured the EPF, we cannot directly determine the offset between seawater and EPF. As mentioned in the discussion (L 528-535) and based on the El/Ca pattern, we propose that pumping into the EPF is most strongly impacting the shells' composition (more than kinetic effects during calcification). Therefore we conclude that it is safe to infer the EPF reaction times from the compositional changes of the shells. However, we refrain from more directly inferring the EPF composition. We now have more clearly stated the fact that we do not have direct EPF composition measurements (L 529-530). Furthermore we have included distribution coefficients between seawater and shell, by directly comparing the two end-members for which quantitative data are available (L 527-528, Table S5).

*2. - Include a comparison between the chemistry and elemental variability of your cultured tridacnid and natural/wild specimens that have been analyzed in the literature.*

We have included a comparison of the El/Ca range to other fossil and recent tridacnids measured with a similar methodological setup, namely LA-ICPMS with a rotatable slit, as presented in Warter and Müller (2018) and Arndt et al. (2025). The comparison is found in the main text (L 321-332) and a supplementary table with the data is added (Table S4).

*3. - Please try to identify the species of tridacnid studied here, possibly with help from the second reviewer.*

Daniel Killam has kindly identified the species for us, and they are included in the sample description (L 90-93). His helpful contribution is acknowledged in L 646-647.

*4. - Identify the measurement of nutrients in (culture) water surrounding tridacnids as an important avenue for further enhancing our understanding of their shell mineralization, metabolism and shell chemistry.*

We state that we did not measure nutrient availability and acknowledge that nutrient measurements would have been an interesting addition to further the understanding of the metabolic activity of giant clams and their photosymbionts (L 427-429). We also state that we assume through constant refreshening of the water around the two clams that we would expect reduced daily nutrient variability compared to a reef environment (L 418-426).

*Some tables contain coloured cells or/and coloured values.*
*Please note that this will not be possible in the final revised version of the paper due to HTML conversion of the paper. When revising the final version, you can use footnotes or italic/bold font.*

We have removed the cell colours from the table.

*Please ensure that the colour schemes used in your maps and charts allow readers with colour vision deficiencies to correctly interpret your findings. Please check your figures using the Coblis – Color Blindness Simulator (https://www.color-blindness.com/coblis-color-blindness-simulator/) and revise the colour schemes accordingly with the next file upload request.*

We ensured that in our figures colour is used to support understanding but is never necessary to interpret the data. The colour schemes have been revised to be better distinguishable for readers with Protanopia, Deuteranopia and Tritanopia, and all information remains clear in monochromatic view.

**RC1 Comments:**

*Figures 3 and 4 are central to the paper. In Figure 3, I would suggest expanding it width-wise. I know Fig. 4 provides an expanded version but the legibility in Fig. 3 is not great and should be improved. This would make the timing of the spiking vs the periodicity in the El/Ca values (which is key) a bit clearer. In Fig. 4, explain the green bars (as you do in Fig. 3) in the figure caption or in the figure.*

We have widened the subplots of Fig. 3 and have added more space around the headings to improve legibility. We have included the explanation of the green bars in the caption of Fig. 4 (L 300), thank you for noticing the missing description.

*In terms of experimental, given that two clams share each jar, and therefore the same water, can they be considered as true replicates? Isn't the carbonate chemistry in the water an important part of the experimental unit?*

The carbonate chemistry is indeed an important part of the experiment. For the two clams grown in the same jar only one water measurement is available. The resulting elemental ratios (El/Ca) data could be expected to be the same, given water chemistry and treatment was identical, if one assumes that only environmental factors impact El/Ca. The observed variability is likely caused by the individual's physiological performance and behaviour. Since the El/Ca data are different between some clams grown in the same culturing jar, we deduce that the individual response of each clam places a limit on overall signal reproducibility. However, most features, like the daily cycles and trends from changing the water source they grew in, are reproducible. The reproducibility between two clams given the same treatment in two different jars is just as good (or as bad) as between two clams grown in the same jar. Nevertheless, we agree that a distinction between replicates grown in the same water and replicates that just grew under the same experimental conditions adds clarity and we have provided this clarification in the revised document (L 305-316).

*Line 109: A few more details about the feeding regime would be good. Quantity? Timing? DOM vs particulate matter? This is important for understanding food vs light contributions.*

The food is from the German brand "fauna marine", and the food powder used is called "coral sprint". It is a fine powder with 85 % protein, 11 % fat, 3% fibre and 1 % ash. It contains additives per 1 kg: 600 i.u. Vitamin D3 (E671), 50 mg iron-sulphate-monohydrate, 2.2 mg calcium-iodate, 6 mg copper-sulphate, 17 mg manganese-monohydrate, 120 mg zinc-monohydrate, 57 mg antioxidants. The recommended dosage is one measurement cup every two days for 500 l of water. We scaled the dosage down to be appropriate for the 11 l water reservoirs attached to the

culturing jars. The clams grew in the prepared reservoir water for the initial spike (experiment days 1-3) and in a new set of reservoir water prepared in the same way for the three days of day and night-spiking (experiment days 4-7). The reservoir water was pumped through the culturing jars continuously, with approximately 2 of the 11 l flowing through the culturing jar every 12 h. There were no feeding events that could have introduce spiked nutrient availability, but the constant pumping of new reservoir water into the jars provided constant nutrient inflow (added to the manuscript L133-134). Through reintegrating the 2 l flown through the culturing jars to the respective 11 l reservoirs every 12 h the nutrient availability within the reservoir water was slightly diluted over the 3 days of usage. Following the feeding instructions, the food within the reservoir water was sufficient throughout the culturing (added to the manuscript L155-156). Before and after the experiment, as well as during the night from experiment day 3 to 4 the clams stayed in a 400 l aquarium with corals, anemones, hermit crabs, fish, sea urchins and other molluscs. In this aquarium 10 ml of "Reef Energy Plus" from the brand "Red Sea" are added daily during the week. We have added information on the feeding in the revised manuscript (L 111-118).

*Consider uncertainties in using 12h ΔALK to estimate calcification.*

This is a good point, thank you. The uncertainties are now included in Table 2 and mentioned in the manuscript (L 233, 237).

*How does the data from this study compare with daily-resolved trace element/Ca data from clams that grew in natural reef settings (e.g. in some of your other publications)? E.g. amplitude, relative phasings. If similar, this would strengthen any arguments you had that your observations apply to Tridacna more generally, not just those with very specific culture conditions. (Juvenile, small individuals in 28 °C, 37 psu, constant light regime (12:12 h) under lab flow are quite different from heterogeneous reef conditions. Whether the 5× daytime calcification and the precise phasing of maxima/minima are similar across sizes, species, and natural diel PAR/temperature cycles remains to be tested – worth acknowledging perhaps?)*

Regarding comparison to recent and fossil *Tridacna*: See answer to the Editors comment 2.

We point out that this particular experiment resulted in this calcification pattern, which might not be uniformly applicable to all species, sizes and temperature regimes in L 319-320.

*The discussion could more explicitly link findings to potential paleoenvironmental applications (ENSO, storm reconstructions, etc.), strengthening the broader impact.*

Thank you for pointing this out. We have strengthened the link to paleoenvironmental applications by adding the discussion chapter 4.6 "Relevance for palaeoclimate applications".

*Language generally fluent, but some sentences are long and could be simplified for clarity.*

We have revised the text for readability, simplicity and clarity and have shortened several sentences that seemed difficult to read.

*Don't capitalise 'tridacnid'*

We have changed this accordingly (L 444, 458, 471).

*Minor typos: Hyphen needed in 'non spiked' on p121. Centre data in table.*

Thank you for noticing, the typo has been corrected (L 121). The data entries in Tables 1 and 2 are now centred.

**Comments RC2:**

*22: I'd advocate for the authors noting here that they are estimating the EPF composition based on the composition of external seawater in a culture setup. This is totally a valid approach, but the abstract made it sound like direct sampling of the EPF was conducted, which does not seem to be the case.*

We have stated more clearly that we estimate the composition of the EPF from that of the shell (L 22), thanks.

*86: It is crucial to know the species of these clams, for reproducibility purposes. I would be happy to personally identify the species if the authors have pictures of the clams. Feel free to email pictures.*

Thank you very much for the identification, which is now included in L 90-92 and Tab. S1.

*109: Do you know the brand/composition of coral food? The diet could be important for other workers to reproduce the results. I think it was useful that the clams were fed (that is, of course different than knowing if they actually ate the food), since other studies often do not feed the clams, which leads to changes in behavior, particularly for juvenile clams of this size, which are generally more reliant on filter-feeding. Did you observe them feeding during the experiment? Production of feces/pseudofeces, valve clapping, etc.*

Please find the details to the food composition and feeding procedure in the comment for Referee #1. We did not directly observe feeding behaviour during the experiment.

*140: The measurement of alkalinity differences at night and day is a great approach and one of those things that makes me slap my forehead, and wonder why someone hadn't thought to do that before, especially since in the aquarium trade, giant clams are known as infamous alkalinity sinks. Were any other environmental parameters measured, even intermittently? I'd be particularly interested in nutrient measurements like nitrate or ammonium (more on that below)*

Sadly, we did not conduct nutrient measurements on the water. We agree that nutrient measurements of the culturing water would be an interesting dataset for future culturing setups and have mentioned this in L 427-431.

*204: What direct influence would the calcification have on DO? I would have thought O2 is more related to photosynthesis. In daylight hours, the clams can be net sources of O2 (Fisher et al., 1985), while at night, they conduct respiration, analogous to similar processes seen in plants. Were there any algae in the tanks with them adding to these processes, or were they the only significant biomass in the observation tanks?*

While the giant clams should consume more oxygen when they are active during the day through respiration, the photosymbionts also produce oxygen during daytime. We clearly see an increase in the oxygen concentration during the day (high calcification rates) and a reduction in oxygen during the night (low calcification rates). That would strengthen the hypothesis that the photosymbiont activity is strongly coupled to the calcification rate. This is now included in the manuscript (L 546-548). The clams (with their symbionts) were the only significant biomass in the tank. Some clams had small amounts of algae still stuck on their shells but that was very little (we did not brush them before transfer into the culturing jars). These algae will have contributed little to the overall oxygen production at daytime.

*336: One source of potential error in estimations of day-night differences in EPF residence time: did the clams partially close at night? They tend to close partially at night (see Killam et al., 2023) in a defensive posture (this is not always the case, such as some aquarium settings where predation is not an issue, so yours might not have). But if they close, their overall internal volume would be smaller, and as such their extrapallial space would be smaller in volume.*

They were not fully closed when the lights were off, but the mantle may have been more extended outward over the ventral shell rim during daytime (light exposure). As we did not monitor valve movement it is difficult to say how this affected the EPF volume. They can retreat their mantles and close quite quickly and that rapid response should not change the EPF volume, although it is logical that if they have less space for several hours, they might reduce the EPF volume as well. We now discuss the EPF volume in L 386-396.

*359: What nutrients are higher in daytime? Generally, N-bearing nutrients would be expected to be lower in the day on average due to assimilation and higher at night due to remineralization/respiration being dominant.*

You are absolutely right, nutrient availability should be higher at nighttime. Thanks a lot for noticing this mistake, we have changed the statement accordingly (L 420-421).

*Section 4.4.1: I suspect that in your tanks, this was not a major factor, but the reason I asked about nutrient measurements is that in other experimental setups where other biota might be present in the aquarium, I believe nitrification would represent a significant source of error trying to replicate this approach. In closed-system aquarium setups, ammonium waste from the inhabitants is converted to nitrate, a process which can destroy alkalinity. In your constrained single-species tanks, I bet that the clams were quickly re-absorbing all ammonium produced to feed their symbionts, making it not a factor of concern in your calculations. However, I'd advise adding a mention here of that process to aid in reproducibility, since this hidden sink of alkalinity could make replication difficult in some research aquaria. Also for that reason, if you have any observations of nutrients like NH4/NO3/NO2/etc during the experiment, please include them.*

As stated above we did unfortunately not measure nutrient contents in the water (now clearly stated in the manuscript L 428). We have included the point that isolated culturing experiments are important for reliable alkalinity measurements (L 125-127).

*376: It is better to refer to them as Tridacninae or tridacnines, since they are in the family Cardiidae*

Thanks for pointing this out. We have changed it (L 385)

Kind regards,

Iris Arndt on behalf of all coauthors